# Characterization of Microorganisms from *Protaetia brevitarsis* Larva Frass

**DOI:** 10.3390/microorganisms10020311

**Published:** 2022-01-28

**Authors:** Huina Xuan, Peiwen Gao, Baohai Du, Lili Geng, Kui Wang, Kun Huang, Jie Zhang, Tianpei Huang, Changlong Shu

**Affiliations:** 1State Key Laboratory of Ecological Pest Control for Fujian and Taiwan Crops, Key Laboratory of Biopesticide and Chemical Biology of Ministry of Education & Ministerial and Provincial Joint Innovation Centre for Safety Production of Cross-Strait Crops & Biopesticide Research Center, College of Life Sciences, Fujian Agriculture and Forestry University, Fuzhou 350002, China; xuanhuina@126.com; 2State Key Laboratory for Biology of Plant Diseases and Insect Pests, Institute of Plant Protection, Chinese Academy of Agricultural Sciences, Beijing 100193, China; 82101195125@caas.cn (P.G.); dubaohai03@163.com (B.D.); llgeng@ippcaas.cn (L.G.); wangkui01@caas.cn (K.W.); zhangjie05@caas.cn (J.Z.); 3Genliduo Bio-Tech Corporation Ltd., Xingtai 054000, China; plaeeihk@163.com

**Keywords:** *Protaetia brevitarsis*, frass, *Bacillus*, identification, bioassay

## Abstract

Decomposers play an important role in the biogeochemical cycle. *Protaetia brevitarsis* larvae (PBLs) can transform wastes into frass rich in humic acid (HA) and microorganisms, which may increase the disease resistance of plants and promote plant growth. Beyond HA, the microorganisms may also contribute to the biostimulant activity. To address this hypothesis, we investigated the potential microbial community in the PBL frass samples and elucidated their functions of disease resistance and plant growth promotion. High-throughput sequencing analysis of four PBL-relevant samples showed that their frass can influence the microbial community of the surrounding environment. Further analysis showed that there were many microorganisms beneficial to agriculture, such as *Bacillus*. Therefore, culturable *Bacillus* microbes were isolated from frass, and 16S rDNA gene analysis showed that *Bacillus subtilis* was the dominant species. In addition, some *Bacillus* microorganisms isolated from the PBL frass had antibacterial activities against pathogenic fungi. The plant growth promotion pot experiment also proved that some strains promote plant growth. In conclusion, this study demonstrated that the microorganisms in the PBL frass are conducive to colonizing the surrounding organic matrix, which will help beneficial microbes to increase the disease resistance of plants and promote plant growth.

## 1. Introduction

In terrestrial ecosystems, the process of decomposition is vital because it produces useful substances that act as fertilizers to enrich soil fertility. The cetoniid beetle *Protaetia brevitarsis* (PB) (Coleoptera: Scarabaeidae) is an easy-to-raise litter-feeding soil insect that can convert decaying plant biomass into frass with a high humic acid (HA) content [1,2,3]. Beyond converting agricultural waste, the mature PB larvae (PBLs) have other beneficial economic effects: (1) the processed mature larva is a traditional medicine that has many benefits for human health [1,4,5,6,7,8], and (2) the mature larva contains high-quality fat and protein, which can be considered a future feed and food source [9]. Therefore, PB and its application have recently become a meaningful research subject. Recent data have shown that PBL frass can be directly applied as a biostimulant without further composting treatment [2,3] and can effectively promote plant growth and improve plant disease resistance [10]. These effects of PBL frass may be largely contributed by its HA [2,3], which has been proven to be an effective plant biostimulant [11,12]. Beyond HA, the microorganisms in PBL frass are also noteworthy factors. First, the intestinal tracts of litter-feeding soil macroinvertebrates have been proven to be favorable habitats for microorganisms and always harbor a dense and active gut microbiota [13,14], and the PBL gut and frass contain a large number of microorganisms [15,16]. However, the extent and importance of such microorganisms and their stimulatory effect on plants are poorly understood.

Currently, the application of beneficial microorganisms to agriculture [17,18] is an important alternative strategy for providing healthy food in a sustainable manner by reducing the amount of chemical fertilizers, chemical pesticides and herbicides used [19,20,21]. Among these commonly used microorganisms (*Azospirillum, Bacillus, Mycorrhizae, Pseudomonas, Rhizobia, Streptomyces* and *Trichoderma*) [22], *Bacillus* species are most promoted for their convenience in production and application. In this study, the bacterial community in the PBL frass converted from spent enoki mushroom substrates were identified to determine the potential impact of frass bacterial communities on agriculture. Then, *Bacillus* strains that are commonly used in the agriculture were isolated and characterized from the community to evaluate the application value of microorganisms in PBL frass.

## 2. Materials and Methods

### 2.1. Insects, Rearing Conditions and Sampling

The PB laboratory population used in this study was originated from a field population collected from Gonzhuling, Jiling Province [1]. Approximately 800 PBLs were reared in a plastic box (65 × 45 × 15.5 cm) at room temperature (25 °C) and fed with enoki mushroom substrates (approximately 50% H_2_O). The control treatments of spent mushroom substrates (SMS) were kept under the same condition. To collect frass, well-grown PBLs were removed and cleaned with sterile water, and then the cleaned PBLs were kept in an empty box for 2 days at 25 °C. The defecated frass was collected for further analysis. At the same time, remaining spent mushroom substrates in PBL feed treatments were also collected. For hindgut samples, the larvae were immersed in 70% alcohol and washed three times with distilled water. Then, the digestive tract of the larva was dissected, and intestines in the hindgut were collected. All the samples were preserved at −80 °C until further analysis.

### 2.2. Bacillus Isolation from PBL Frass

Approximately 1 g of collected frass was suspended in a 50 mL centrifuge tube with 10 mL of sterile water. After incubating in an 80 °C water bath for 30 min, the suspension was serially diluted and spread on LB agar plates, and then the plates were cultured at 28 °C for 24 h. *Bacillus*-like colonies were picked for purification culture. The pure bacterial cultures were stored in 20% (*v*/*v*) glycerol and kept at −80 °C until further use.

### 2.3. High-Throughput Sequencing (HTS) and Bioinformatic Analyses

Before HTS, DNA was extracted by a modified protocol using an AxyPrep Multisource Genomic DNA Miniprep Kit. First, the crushed SMS, fresh frass, hindgut contents and collected bacterial cells were suspended and lysed in a 4 M guanidine isothiocyanate solution for 2 min, respectively. After centrifugation for 5 min at 12,000× *g*, 500 μL of supernatant was transferred to an adsorption column. The remaining processes were carried out according to the manufacturer’s protocol, and the DNA yield was used for bacterial community sequencing or genomic sequencing.

Bacterial community composition was analyzed by amplicon sequencing of the 16S rRNA gene V4 hypervariable region with the universal primer set 515F/806R [23]. PCR was carried out as follows: initial denaturation at 95 °C for 10 min, 35 cycles at 94 °C for 1 min, 55 °C for 1 min, and 72 °C for 1 min, and final extension at 72 °C for 10 min. PCR products were purified using an AxyPrep DNA Gel Extraction Kit (Axygen, Wujiang, China). The purified PCR products were extended with Illumina-specific adaptors using a TruSeq^®^ DNA PCR-Free Sample Preparation Kit (San Diego, CA, USA), and the resulting libraries were sequenced using the Illumina HiSeq 2500 system (2 × 250 bp). After filtering the low-quality reads, Illumina paired-end reads were error-corrected and merged by PANDAseq [24]. Chimeric sequences were eliminated by using the UCHIME “Gold” database, producing high-quality 16S rRNA gene sequences [25]. Then, an OTU table (97% identity was set as the threshold value) was generated and quantified by UPARSE (Usearch version 8.0.1517) [26]. The highest frequency OTUs were selected as representative OTU sequences according to the algorithm principle. Finally, the representative OTU sequences were annotated by the Ribosomal Database Project (RDP version 2.2) classifier Greengenes (version 13.8) [27], and all processes were performed under default parameters.

For bacterial genome sequencing, the 350 bp HTS library was constructed using a TruSeq DNA PCR-Free Library Prep Kit, and sequencing was again performed on an Illumina HiSeq 2500 sequencer (Illumina, San Diego, CA, USA). The produced 150 bp paired-end raw reads were trimmed to remove adaptor sequences, low-quality reads and low-quality bases using Trimmomatic (version 0.38) [28]. Then, the isolate genomes were assembled using SPAdes (version 3.15.3) [29], and the coding gene was predicted by GeneMark [30] with heuristic models. Whole genome-based diversity was represented by phylogenetic trees constructed by CVTree [31] using a composition vector (CV) approach. The secondary metabolite biosynthesis gene clusters from the genome were identified by antiSMASH [32].

### 2.4. Bioassay

To evaluate the potential of isolates against plant fungal pathogens, *Sclerotium rolfsii*, *Fusarium oxysporum* and *Sclerotinia sclerotiorum* were used for confrontation cultures [33]. The confrontation cultures were carried out in 90 mm diameter potato dextrose agar (PDA) plates. A pair of symmetrical wells, which were 30 mm away from a center well, were made. One hundred microliters of each isolate culture, which had been incubated at 30 °C with shaking at 230 rpm for 48 h, was added to two symmetrical wells, while the fungal pathogens were inoculated into the center position. The plates were incubated at 26 °C for 48 h. The observable inhibition zones were used as indicators of the antifungal activities of the isolates.

To evaluate the plant growth promotion effect, greenhouse pot tests were carried out using *Brassica campestris*. The seeds were germinated on moist filter paper at 26 °C for 36 h, reaching a root length between 1 and 2 cm. The germinated seeds were then soaked in the vegetative growth stage bacterial culture (OD_600_ = 1.0) for 20 min to inoculate with bacteria. The inoculated seedling was planted into a square pot filled with a moistened substrate that consisted of a 2:1 ratio of river sand and vermiculite. Each pot had two seedlings and was then placed in a seedling tray cultivated at room temperature for four weeks. The pots with two successfully transplanted plants were used for further analysis. A *t*-test was performed to determine whether there is a significant difference between the treatments and the blank control.

## 3. Results

### 3.1. Microbial Diversity Comparisons between Samples

The bacterial composition of different samples was determined by sequencing analysis of the 16S rRNA gene. After trimming the low-quality regions and removing the short reads and chimeras, a total of 1,320,200 effective tags were identified, with an average length of 416 bp (Table 1). Then, the effective sequences were grouped at 97% DNA sequence similarity, and a total of 4371 OTUs were produced. The rarefaction curve applied to OTUs showed that all samples reached a plateau, suggesting that all samples were sufficiently sequenced to represent their identity. On average, there were 2665.60, 2978.83, 2736.50 and 1595.20 OTUs obtained from the frass (F), larva hindgut (LH), remaining substrate (RS) and substrate control sample (SC9, samples obtained on the ninth day) (Table 1). Among 3869 OTUs identified in the RS samples, 830 OTUs (21.45%) were not detected in the SC9 samples, most of which were detected in the F or LH samples.

The diversity of the microbial population from different samples, based on richness and evenness, was represented by alpha diversity analysis. The alpha diversity patterns varied across the samples. The OTU, Chao1 and Shannon-2 indexes in the F, LH and RS samples were significantly higher than those in the SC9 samples (*p* < 0.01), indicating that the F, LH and RS samples had greater microbial diversity (Table 1). From the samples, 29 phyla, 178 families and 326 genera were identified. At the phylum level, *Proteobacteria* and *Bacteroidetes* were the two most dominant phyla, comprising more than 60% of all detected microorganisms, and these phyla are typically observed in soil libraries. Compared with those in the SC9 samples, the *Firmicutes*, *Elusimicrobia* and *Spirochaetes* abundances increased significantly in the RS samples, in which they were dominant microorganisms (Figure 1A). Both cluster analysis and principal component analysis (PCA) indicated that the microbiota in SC9 samples was clearly separated from that in other samples. In the PCA, PC1 and PC2 explained 56.3% and 18.2% of the global variation, respectively (Figure 1B). In PC1, the SC9 samples were separated from the LH, F and RS samples, while the LH samples were separated from the F and RS samples in PC2. However, the F and RS samples could not be separated from each other in both PC1 and PC2.

At the genus level (Figure 2A), the *Pseudoxanthomonas*, *Cellvibrio*, *Parapedobacter*, *Chitinophaga* and *Paenibacillus* were dominant in SC9 and accumulated an abundance of 14.00%. After PBL conversion, the *Cellvibrio*, *Ohtaekwangia*, *Luteimonas*, *Pseudoxanthomonas* and *Saccharibacteria_genera_incertae_sedis* were dominant in F with a cumulative abundance of 14.80%. Different from SC9, with the effect of PBL and F, the dominant genera in RS were *Prevotella*, *Cellvibrio*, *Haemophilus*, *Microbacterium* and *Saccharibacteria_genera_incertae_sedis*, which accumulated to 10.63% of the bacteria community. In addition, the abundance of beneficial microorganisms commonly used in agriculture were calculated (Figure 2B). *Bacillus*, *Pseudomonas* and *Streptomyces* abundances in F were higher than the upper quartile of 0.05% and consisted of 0.06% (standard deviation: 0.02%), 0.65% (standard deviation: 0.30%) and 0.10% (standard deviation: 0.07%) of the bacterial community, respectively.

### 3.2. Bacillus Isolation and Genome Analysis

From frass samples, 13 *Bacillus* isolates with different colony morphologies were obtained. 16S ribosomal DNA data showed that they belonged to *Bacillus subtilis* or closely related species. To obtain better identification results and analyze the characteristics of these isolates, the draft genomes of these isolates were further determined by HTS. After filtering low-quality reads, each strain produced enough clean data, and the clean reads were subsequently deposited in the Sequence Read Archive of GenBank (Table 2). After assembly by the SPAdes pipeline using default parameters, GeneMark was employed to predict the coding genes from the resulting contigs. The data showed that the genome size of these isolates was between 4.06 and 4.96 Mb, coding for 2876 to 3780 genes.

Once the predicted protein sequences were generated, whole genome-based phylogenetic trees of these isolates were constructed using a composition vector (CV) approach in CVTree3. By comparison with reported species, the thirteen isolates were divided into four species, including one *Bacillus amyloliquefaciens*, five *B**. halotolerans*, two *B. tequilensis* and five *B. subtilis* (Figure 3). To distinguish the isolates at the strain level, we compared the distance between these isolates, and then the smallest distance (0.05) between distinct strains (*B. subtilis* 168_01 and isolate CFG13) was used as the threshold. According to this threshold, the 13 isolates could be divided into 10 groups. Among the groups, the isolates CFG6 and CFG11 (distance 0.03), CFG3 and CFG10 (distance 0.02) and YMCF3 and CFG13 (distance 0.04) with the closest genetic distances were assigned to three groups.

To analyze application potential in agriculture, the secondary metabolite biosynthesis gene clusters in the genomes of the 13 isolates were annotated by the antiSMASH 6.0 pipeline. In total, 108 clusters were identified, 68 of which had high identity (>70%) to the reference cluster. The annotation results indicated that the isolates may have bacillaene, bacillibactin, bacilysin, difficidin, fengycin, macrolactin H, mersacidin, subtilin, sporulation killing factor and subtilosin A biosynthetic potential. The distribution of the high-identity clusters showed that the bacilysin, bacillaene and bacillibactin biosynthetic gene clusters were common in the isolates, while the difficidin, macrolactin H and mersacidin biosynthetic gene clusters were present only in isolate PGCF23 (Table 3).

### 3.3. Bacillus Is a Key Factor for Disease Resistance and Plant Growth Promotion

The antifungal effects of the thirteen isolates against three important plant pathogens were subsequently assessed by confrontation culture analysis. The data showed that all the isolates had antifungal effects on *S. sclerotiorum*, but only seven isolates inhibited *S. rolfsii* and eleven isolates showed antifungal effects against *F. oxysporum* (Figure 4). The isolates CFG2, CFG10, CFG11, CFG13, CFS2 and YMCF3 showed antifungal effects against all plant pathogens. The antifungal ability of isolates CFG2, CFG6, CFS2 and CFS6 was stronger against *S. sclerotiorum*, while that of CFG13, CFS2, CFS5 and YMCF3 was stronger against *S. rolfsii*.

The effects of the *Bacillus* isolates on the plant growth promotion of *B. campestris* were evaluated through greenhouse pot tests. The results indicated that the isolates CFG3, CFG10, CFG11, CFG13 and YMCF3 could significantly promote *B. campestris* growth (Figure 5).

## 4. Discussion

Soil macrofauna are very important for the carbon cycle in terrestrial ecosystems due to their contributions to the depolymerization and fermentative breakdown of the cellulosic or lignocellulosic component of biomass [14]. The digestive tract is a microorganism-favorable habitat that is always enriched in a dense and active gut microbiota that has the ability to degrade organic matter [13]. When the macrofauna defecate, the decomposed organic matter combined with the gut microorganisms are released to the soil ecosystem. The effects of gut microorganisms on ecosystems have attracted researchers’ interest and have been extensively investigated [13,34,35]. However, few studies have focused on the application of these microorganisms in agriculture. In China, as a resource insect, PBL has been applied to convert plant residue and SMS to produce frass with biostimulant activity [2,10]. In this study, beyond HA, which has been proven to be an efficient plant biostimulant [11,12], the potential beneficial contribution of frass bacteria to the agriculture was investigated, which would be of great help for further understanding of the characteristics of insects’ frass and providing guidance for their further application.

Microorganisms from litter-feeding insects may easily colonize litter. Previous data have suggested that the bacterial symbionts enriched in the digestive tract largely contribute to lignocellulose degradation [13] and that soil macroarthropod activity can increase the nitrogen mineralization rate via interactions with microorganisms [34]. In this paper, it was found that the bacteria in frass can colonize the organic matter from which they originated, making the microbial community of organic matter close to that of insect frass (Figure 1). The reason may be that the microbial decomposers are enriched by favorable habitats in the macroarthropod digestive tract, and it is thus easier to gain advantages with regard to subsequent contact with organic matter. In China, researchers use PBLs to transform crop stalks. When the produced insect frass is applied to cultivated land, the microorganisms in insect frass will have the opportunity to colonize the remaining crop stalks in the soil and may accelerate decomposition and even have a chance to act on the roots of subsequent crops.

Based on the taxonomy of the frass bacterial community, after transformation of the PBL digestive system, although microbial diversity increased significantly, there were seven genera, including *Ancylobacter*, *Catonella*, *Coprococcus*, *Fusicatenibacter*, *Lachnoanaerobaculum*, *Pseudoclavibacter* and *Yinghuangia,* that disappeared from the microbial community. In addition, the abundance of genera *Acetobacter*, *Alcaligenes*, *Brevibacterium*, *Chitinophaga*, *Cohnella*, *Neobacillus*, *Ornithinibacillus*, *Paenalcaligenes*, *Pseudopedobacter* and *Stenotrophomonas* decreased by more than 90%. Nonetheless, the genera *Bacillus*, *Pseudomonas* and *Streptomyces* in frass maintained a relatively high abundance (Figure 2). Many species from these three genera are beneficial to agriculture [22], which suggested that PBL frass bacteria may play a positive role in agricultural production.

Currently, many *Bacillus* species are important and beneficial agricultural microorganisms, and the products developed from these *Bacillus* species are widely applied as biological insecticides, fungicides, fertilizers and biostimulants [21,36,37]. In this report, 13 *Bacillus* strains were isolated according to the difference in colony morphology from the PBL frass from enoki SMSs, and the bioassay data showed that many isolates of these strains have good antifungal and growth promotion activities. In addition, the isolates’ genome data were analyzed in this investigation, and the variation of the genome size indicated a great species diversity. According to the genetic distance calculated from the genome data, these isolates were divided into ten groups and assigned to four *Bacillus* species by phylogenetic analysis. The genetic distance calculated from genome level not only provides an accurate taxonomy, but can also distinguish the differences between isolates from same species; thus, the genome-based phylogenetic analysis represented the great *Bacillus* diversity in the frass. Genome data not only provide the information for genetic distance analysis, but genome annotation information can also help researchers to analyze the phenotypic or functional differences. The metabolites synthesized in *Bacillus* species are important for the antifungal and growth-promoting activities. In this paper, the identified secondary metabolite biosynthesis gene clusters in the isolates were rich and diverse (Table 3), which implied a functional diversity. Therefore, the PBL frass contained abundant and valuable *Bacillus* resources, meaning it has good research and application value.

In summary, this study illustrated that in addition to the high content of HA in frass, the bacterial community also contributes to its beneficial acceleration of organic matter decomposition, suppression of plant diseases and promotion of plant growth. In addition, these data further demonstrated the application potential and value of PBLs in circular agriculture.

## Figures and Tables

**Figure 1 microorganisms-10-00311-f001:**
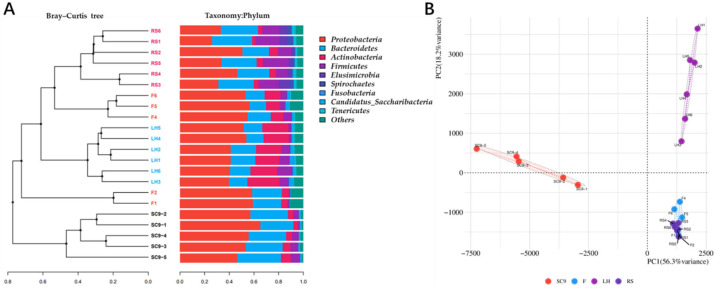
Beta diversity analysis of the samples. (**A**) The Bray–Curtis tree constructed based on the cluster using the unweighted pair-group method with arithmetic means (UPGMA). (**B**) The PCA of the variance between samples.

**Figure 2 microorganisms-10-00311-f002:**
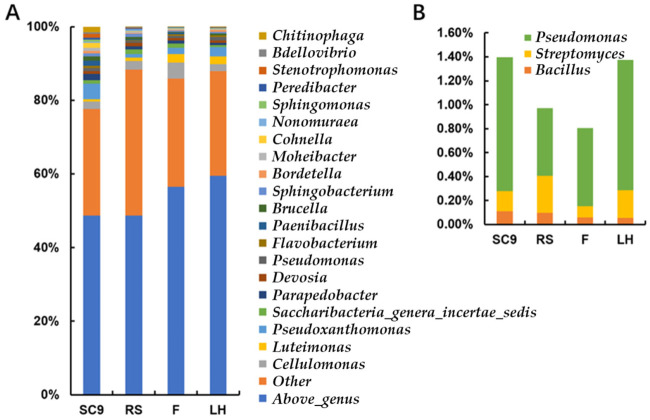
Genus-level abundance of the samples. (**A**) The information for the dominant genera of the samples. “Above_genus” means that the OTU sequence has low identity when compared to the existing genera sequences, and cannot be assigned to the existing genera. (**B**) The abundance of *Bacillus*, *Pseudomonas* and *Streptomyces* in the samples.

**Figure 3 microorganisms-10-00311-f003:**
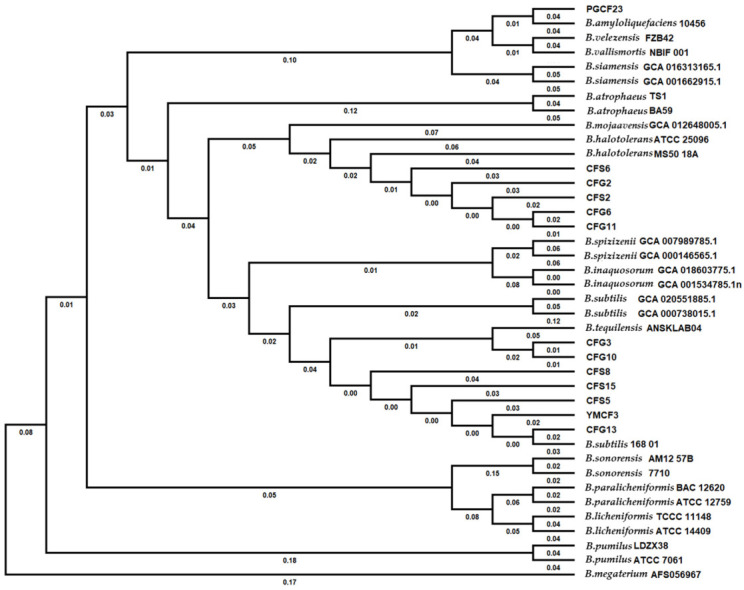
Whole-genome-based phylogenetic tree constructed by the composition vector approach. The distances are shown under the branches.

**Figure 4 microorganisms-10-00311-f004:**
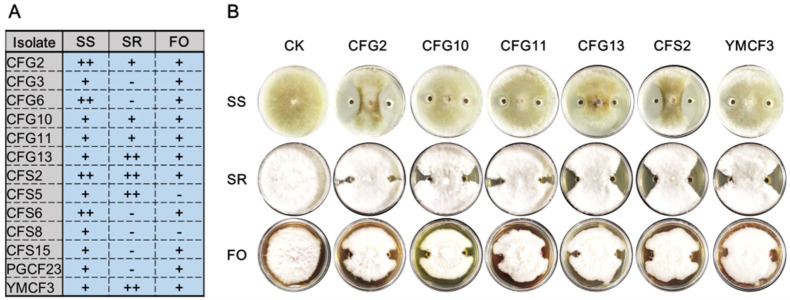
The antifungal effects of the 13 *Bacillus* isolates. (**A**) Evaluation of the antifungal abilities of *Bacillus* against plant pathogens. “−” means no antifungal effect, “+” means weak antifungal effect and “++” means stronger antifungal effect. (**B**) Confrontation culture picture of *Bacillus* against plant pathogens. SS means *S. sclerotiorum*, SR means *S. rolfsii* and FO means *F. oxysporum*.

**Figure 5 microorganisms-10-00311-f005:**
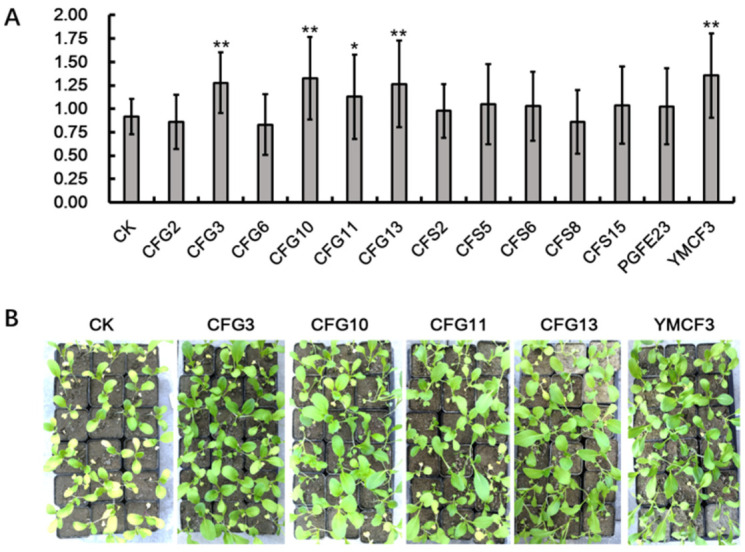
The effects of the *Bacillus* isolates on the plant growth promotion of *B. campestris,* evaluated through greenhouse pot tests. (**A**) Column chart of the plants’ fresh weight. The *Y*-axis represents the weight of plants (g), and error bars indicate the standard deviation. Treatment marked with “**” means *p* < 0.01, while “*” means *p* < 0.05. (**B**) The treatments with a significant plant growth promotion effect. CK means the control treatment.

**Table 1 microorganisms-10-00311-t001:** Summary of samples’ OTU and alpha diversity index.

Sample	OTU		Chao1		Shannon-2		Simpson	
	Average	SD	Average	SD	Average	SD	Average	SD
SC9	1595.20	312.41	1927.32	465.45	7.26	0.22	0.0357	0.0147
RS	2736.50	223.67	3048.20	203.78	9.29	0.12	0.0058	0.0006
F	2665.60	246.70	3053.10	260.38	8.56	0.22	0.0093	0.0014
LH	2978.83	195.02	3296.92	158.72	8.93	0.31	0.0119	0.0045

Note: the SD means standard deviation.

**Table 2 microorganisms-10-00311-t002:** Summary of sequenced reads and assembled contigs of *Bacillus*.

Isolate	Reads		Contigs		
	Accession No.	Clean Data (Gb)	Total Bases (Mb)	N50 Length (Mb)	CDs No.
CFG2	SRR16943721	1.83	4.96	0.23	3230
CFG3	SRR16943720	1.66	4.18	1.07	2948
CFG6	SRR16943709	0.98	4.24	0.36	3169
CFG10	SRR16943708	2.16	4.27	0.89	3010
CFG11	SRR16943707	0.98	4.19	0.36	3097
CFG13	SRR16943706	2.13	4.78	1.02	3780
CFS2	SRR16943705	1.07	4.26	0.50	3071
CFS5	SRR16943704	1.17	4.10	1.02	2876
CFS6	SRR16943703	1.98	4.92	0.37	3148
CFS8	SRR16943719	1.04	4.12	0.99	3050
CFS15	SRR16943718	1.04	4.06	1.02	2965
PGCF23	SRR16943713	1.02	4.16	0.62	2972
YMCF3	SRR16943711	0.98	4.07	1.02	2984

**Table 3 microorganisms-10-00311-t003:** Summary of secondary metabolite biosynthesis gene clusters in the isolates.

Cluster	CFG2	CFG3	CFG6	CFG10	CFG11	CFG13	CFS2	CFS5	CFS6	CFS8	CFS15	PGCF23	YMCF3
Bacilysin	100	100	100	100	100	100	100	100	100	100	100	100	100
Subtilosin A	100	100	100	100	100	100	100	100	100	100	100		100
Bacillaene	100	100	100	100	100	100	100	100	100	100	100	100	100
Bacillibactin	84	100	84	100	84	100	100	100	100	100	100	100	100
Fengycin	80	73	80	73	80	86	80	86	73	80	73		80
Subtilin		100		100									
SKF						100		100					100
Difficidin												100	
Macrolactin H												100	
Mersacidin												100	

Note: SKF means sporulation killing factor. The numbers listed in the table are the percentages of the identity of the secondary metabolite biosynthesis gene cluster in the isolate genome compared with the references.

## Data Availability

Genome sequencing reads are deposited in the Sequence Read Archive of GenBank.

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
