# Peer review of "Characterization of Microorganisms from Protaetia brevitarsis Larva Frass"

_microorganisms, 2022, doi:10.3390/microorganisms10020311_

Round 1

Reviewer 1 Report

Everything is well explained except the part for the growth promotion of Brassica campestris.

That part of the method lacks experimental design, which statistical model did you choose and which statistical analysis did you perform? From the results, I assume that you did the Anova analysis, but I don't see which post hoc analysis you did. I also don't see which program you used for statistical analysis or did you calculate manually? You should write that.

Best regards

Author Response

Many thanks to the reviewers for suggestions, we seriously revised and marked them in green fonts in the article.

Reviewer's comments:

Reviewer 1:

Comments and Suggestions for Authors

Everything is well explained except the part for the growth promotion of Brassica campestris.

That part of the method lacks experimental design, which statistical model did you choose and which statistical analysis did you perform? From the results, I assume that you did the Anova analysis, but I don't see which post hoc analysis you did. I also don't see which program you used for statistical analysis or did you calculate manually? You should write that.

My response:

We do not want to compare growth-promoting activity between isolates, and only want to determine if there is a significant difference between the means of the treatment and the blank control. Therefore, a t-test was employed to perform statistical analysis. The introduction was added in line 132-133.

Reviewer 2 Report

This study investigated the diversity and function of microbial communities associated with larva frass of Protaetia brevitarsis. The community of microbiome was compared between samples and significant differences were found. Bacillus strains were isolated and its genome sequences were obtained. Some strains showed a significant increase in plant growth, which suggests a beneficial function of bacteria in the larva fass of Protaetia brevitarsis. The subject and process of this study are sufficient to be recommended for publication. However, several points are needed to be revised.

  1. The metabarcoding results cannot be interpreted well because abbreviations of samples (RS, LH, F, SC9) were used without the introduction of full names. Also, detailed results such as genus-level abundance were not presented. In lines 166-170, the abundance of Bacillus, Pseudomonas and Streptomyces were described, but I cannot know this is sufficiently high in frass samples compared to other samples. Also, the information for the dominant genera is lacking in the results.

  1. In the Discussion, only the general information and beneficial effect of microorganisms were discussed in this part. The results of metabarcoding and genome analysis were not discussed well; the authors obtained many results from these analyses, but the meaning of these results is not presented.

Minor points:

Please use italics for “Bacillus” (e.g. L78, L272).

Line 53: Remove Mycorrhizae.

L99: The capital letter, UPARSE.

L166: Add a space before [33].

L272: remove “resources”.

Table 3: what is the meaning of the values? Identity? Why identity is important?

Author Response

Many thanks to the reviewers for suggestions, we seriously revised and marked them in green fonts in the article.

Reviewer's comments:

Reviewer 2:

Comments and Suggestions for Authors

This study investigated the diversity and function of microbial communities associated with larva frass of Protaetia brevitarsis. The community of microbiome was compared between samples and significant differences were found. Bacillus strains were isolated and its genome sequences were obtained. Some strains showed a significant increase in plant growth, which suggests a beneficial function of bacteria in the larva fass of Protaetia brevitarsis. The subject and process of this study are sufficient to be recommended for publication. However, several points are needed to be revised.

The metabarcoding results cannot be interpreted well because abbreviations of samples (RS, LH, F, SC9) were used without the introduction of full names. Also, detailed results such as genus-level abundance were not presented. In lines 166-170, the abundance of Bacillus, Pseudomonas and Streptomyces were described, but I cannot know this is sufficiently high in frass samples compared to other samples. Also, the information for the dominant genera is lacking in the results.

 My response:

We add introduction of full names for RS, LH, F, SC9 in line 143-144. we add a column chart to present the genus-level abundance as well as the abundance of Bacillus, Pseudomonas and Streptomyces compared to other samples. The description is added in line 168-174.

In the Discussion, only the general information and beneficial effect of microorganisms were discussed in this part. The results of metabarcoding and genome analysis were not discussed well; the authors obtained many results from these analyses, but the meaning of these results is not presented.

My response:

We have revised it as reviewer’s suggestion, added sentences in line 288- 301, and marked it in green text.

Minor points:

Please use italics for “Bacillus” (e.g. L78, L272).

My response:

We have revised it as reviewer’s suggestion and marked it in green text.

Line 53: Remove Mycorrhizae.

My response:

We have revised it as reviewer’s suggestion.

L99: The capital letter, UPARSE.

My response:

We have revised it as reviewer’s suggestion and marked it in green text.

L166: Add a space before [33].

My response:

We have revised it as reviewer’s suggestion.

L272: remove “resources”.

My response:

We have revised it as reviewer’s suggestion.

Table 3: what is the meaning of the values? Identity? Why identity is important?

My response:

The values are the identity of the secondary metabolite biosynthesis gene cluster in the isolate genome comparing with the references. We used 70% as threshold to screen out the reliable alignment results. Because multiple genes in the cluster are required to synthesize effective secondary metabolites, thus, a relatively high threshold is selected here. The identity data presented here not only showing the existence of the corresponding gene cluster, but also providing the consistency data between the corresponding gene cluster and the reference sequence.

We add the instruction of the value in line 219 -220.

Round 2

Reviewer 2 Report

The authors have revised the manuscript thoroughly, and the manuscript is ready to be published after minor revision as follows:

L143: Give a space after “frass”.

L144: I think “CK” is not needed. Remove it or change “substrate CK” to “control substrate” or “substrate control sample” or others.

L169, 171, 173, 275: Change “was” to “were”.

L170, 172, 174, 177, 178: Write only to the first or second decimal place (e.g. 14.0000% to 14.0%).

L278-279: I don’t think the abundance of Bacillus, Pseudomonas, and Streptomyces were high; the sum of these genera is only about 1%!

Figure 2A: What is the meaning of “Above_genus”? Please add the meaning to the legend.

Author Response

Many thanks to the reviewers for suggestions, we seriously revised and marked them in green fonts in the article.

Reviewer's comments:

Reviewer 2:

Comments and Suggestions for Authors

The authors have revised the manuscript thoroughly, and the manuscript is ready to be published after minor revision as follows:

L143: Give a space after “frass”.

My response:

We have revised it as reviewer’s suggestion.

L144: I think “CK” is not needed. Remove it or change “substrate CK” to “control substrate” or “substrate control sample” or others.

My response:

We have revised it as reviewer’s suggestion and marked it in green text.

L169, 171, 173, 275: Change “was” to “were”.

My response:

We have revised it as reviewer’s suggestion and marked it in green text.

L170, 172, 174, 177, 178: Write only to the first or second decimal place (e.g. 14.0000% to 14.0%).

My response:

We have revised it as reviewer’s suggestion and marked it in green text.

L278-279: I don’t think the abundance of Bacillus, Pseudomonas, and Streptomyces were high; the sum of these genera is only about 1%!

My response:

Indeed, the microbial abundances of these genera were relatively low compared to the dominant genera.

However, considering the number of genera (307) that can be detected in the frass, the abundance of Bacillus, Pseudomonas, and Streptomyces were not low. And, this doesn't even include a ton of Above_genus OTUs. Only consider OTUs that can be attributed to the genus level, ranked by abundance, the rankings of Bacillus, Pseudomonas, and Streptomyces are 75th, 14th, and 54th respectively. Therefore, we believed that abundances of these genera were relatively high, higher than the upper quartile of 0.0525%.

Figure 2A: What is the meaning of “Above_genus”? Please add the meaning to the legend.

My response:

“Above_genus” means that the OUT’s sequence with low identity when compared to the existing genera sequences, and cannot be assigned to the existing genera. Usually, these OTUs were from some unreported new species. We added the meaning to the legend and marked it in green text.